# 2-Methoxyestradiol-3,17-*O*,*O*-bis-sulfamate (STX140) Inhibits Proliferation and Invasion via Senescence Pathway Induction in Human BRAFi-Resistant Melanoma Cells

**DOI:** 10.3390/ijms241411314

**Published:** 2023-07-11

**Authors:** Ylana Adami Franco, Manoel Oliveira de Moraes, Larissa A. C. Carvalho, Wolfgang Dohle, Renaira Oliveira da Silva, Isabella Harumi Yonehara Noma, Keli Lima, Barry V. L. Potter, João A. Machado-Neto, Silvya Stuchi Maria-Engler

**Affiliations:** 1Department of Clinical and Toxicological Analyses, School of Pharmaceutical Sciences, University of São Paulo, Avenida Professor Lineu Prestes, Butantã 05508-000, São Paulo, Brazil; ylana.adami@gmail.com (Y.A.F.); manoelooliveira94@gmail.com (M.O.d.M.J.); lari.anastacio@gmail.com (L.A.C.C.); oliveirarenaira@usp.br (R.O.d.S.); isabellanoma@usp.br (I.H.Y.N.); 2Medicinal Chemistry & Drug Discovery, Department of Pharmacology, University of Oxford, Mansfield Road, Oxford OX1 3QT, UK; wolfgang.dohle@pharm.ox.ac.uk (W.D.); barry.potter@pharm.ox.ac.uk (B.V.L.P.); 3Department of Pharmacology, Biomedical Sciences Institute, University of São Paulo, Avenida Professor Lineu Prestes, Butantã 05508-000, São Paulo, Brazil; kelilima@usp.br (K.L.); jamachadoneto@usp.br (J.A.M.-N.)

**Keywords:** STX140, estradiol analogue, melanoma, senescence, B-raf-inhibitor resistance

## Abstract

The endogenous estradiol derivative 2-Methoxyestradiol (2-ME) has shown good and wide anticancer activity but suffers from poor oral bioavailability and extensive metabolic conjugation. However, its sulfamoylated derivative, 2-methoxyestradiol-3,17-*O*,*O*-bis-sulfamate (STX140), has superior potential as a therapeutic agent, acts by disrupting microtubule polymerization, leading to cell cycle arrest and apoptosis in cancer cells and possesses much better pharmaceutical properties. This study investigated the antiproliferative and anti-invasive activities of STX140 in both SKMEL-28 naïve melanoma (SKMEL28-P) cells and resistant melanoma cells (SKMEL-28R). STX140 inhibited cell proliferation in the nanomolar range while having a less pronounced effect on human melanocytes. Additionally, STX140 induced cell cycle arrest in the G2/M phase and sub-G1, reduced migration, and clonogenic potential in monolayer models, and inhibited invasion in a 3D human skin model with melanoma cells. Furthermore, STX140 induced senescence features in melanoma and activated the senescence machinery by upregulating the expression of senescence genes and proteins related to senescence signaling. These findings suggest that STX140 may hold potential as a therapeutic agent for melanoma treatment.

## 1. Introduction

Melanoma is one of the most aggressive types of cancer that arises from melanin-producing cells, the melanocytes [1]. Although it has a low incidence rate compared to other types of skin cancers, it manifests a relatively high mortality number representing a huge public health problem throughout the world [2,3]. The worst prognosis given to patients with melanoma is mainly due to its typical characteristic of evasion of the immune system and the high rate of cell proliferation and metastasis [4,5].

Melanoma’s high proliferative rate arises mainly from alteration in the MAPK (Mitogen-Activated Protein Kinase) pathway in around 90% of patients, and essentially, the BRAF^V600E^ point mutation represents about 50% of cutaneous melanoma [6,7,8]. Thus, tumor cells with the BRAF^V600E^ mutation usually maintain the active conformation of this intracellular receptor, constitutively activating MAPK signaling, which is responsible for stimulating cell survival, proliferation, migration, apoptosis evasion, and metastasis [9,10]. Therefore, the MAPK pathway is currently the primary target in melanoma treatment, mainly with BRAF-inhibitors (BRAFi), such as vemurafenib [11,12].

Although the development of BRAFi has revolutionized melanoma treatment increasing overall survival (OS, 14–17 months) and response rate (RR, 48–53%) over dacarbazine (5.6–7.8 m; 5–20%, respectively), it is expected that in a 6- to 12-month period, patients experience a therapeutic relapse due heterogeneity that gives rise to intrinsic and acquired resistant cells that repopulate the tumor [13,14]. Furthermore, among the various resistance phenotypes, the hyperactivation of compensatory pathways such as PI3K-AKT-mTOR, overexpression of tyrosine kinase receptors, BRAF/CRAF isoform switch, and metabolic reprogramming are some challenges [15]. More recently, clinical trials demonstrated that polytherapy with BRAFi associated with MEK inhibitors and immune checkpoint modulators could increase OS (29.8 months) and response rate (63%), postponing resistance phenotype appearance [5,16]. Despite these recent efforts, new therapeutic alternatives aiming to overcome BRAFi resistance mechanisms that target metastatic melanoma are still an open public health demand [11].

In this context, a major endogenous metabolite of 17-beta-estradiol, 2-methoxyestradiol (2-ME), has been studied in vitro, in vivo, and in clinical trials against several types of cancer, including melanoma. Our previous study reported that 2-ME activity can induce cell death, it acts as an antiangiogenic compound and as a senescence inductor [17]. In addition, other studies showed that 2-ME also reversed metabolic-related pro-carcinogenic phenotype in osteosarcoma cells [18], inhibited skin injuries induced by radiation exposure [19], increased the infiltration of cytotoxic CD8+ T lymphocytes and the expression of PD-L1 in B16 tumors, in which authors suggest enhancing the efficacy of PD-1 blockade immunotherapy [20]. However, despite promising results, 2-ME has shown limited application due to its poor oral bioavailability (<2%) and extensive bioconjugation in vivo [21,22].

Therewith, the 2-ME derivative, 2-methoxyestradiol-3,17-*O*,*O*-bis-sulfamate (commonly referred to as STX140, Figure 1) has demonstrated excellent oral bioavailability (>85%) [23,24,25,26]. In addition, STX140 also demonstrated an estradiol-independent induction of angiogenesis, along with an in vitro and in vivo multitargeting anticancer activity, including breast and prostate cancer models [27,28,29,30]. Recently, STX140 was shown to have excellent in vivo activity in an animal model of multiple sclerosis [31]. However, STX140’s potential action against melanoma model cells, especially those with the resistant BRAFi phenotype, still needs to be addressed. Therefore, the present work analyzes the senescence-inducing activity and antiproliferative and anti-invasive effects of STX140 against metastatic melanoma with BRAF mutation and BRAFi-resistant cells.

## 2. Results

### 2.1. STX140 Showed Antiproliferative and Anti-Invasive Activity in Melanoma Cells

As evidenced in the curves of Figure 2a and Table 1, STX140 demonstrated cytotoxicity in the nanomolar range for all cell lines tested, being non-selective for resistant or parental melanomas and having a less pronounced effect on melanocytes (IC_50_ = 222.3 nM). The compound potency against melanoma cells increased time-dependently around 30–40%, but only in 72 h-incubation (61.9 and 68.2 nM for SKMEL-28P and -28R, respectively), with no pronounced variation in IC_50_ between 24 h (95.3 and 112.3 nM for SKMEL-28P and -28R, respectively) and 48 h (114.9 and 101.0 nM for SKMEL-28P and -28R, respectively). Considering prolonged and chronic exposure to STX140, the clonogenic evaluation showed inhibition of the clonogenicity potential of resistant melanoma from 50 nM up and for both melanoma (parental and resistant) cells from 100 nM, with total clonogenic inhibition at 200 nM.

In addition, STX140 (100 nM) also demonstrated a significant ability to inhibit cell migration after 24 and 48 h of incubation, with a more pronounced effect for SKMEL-28P in the monolayer model (Figure 3). In the three-dimensional skin model (Figure 4), STX140 (500 nM) demonstrated cytotoxicity with inhibition of SKMEL-28P proliferation when compared to the untreated control, in addition to inhibition of invasion of the dermal layer by SKMEL28-R compared to the untreated control. However, STX140 treatment did not appear to affect tissue formation when evaluating non-melanoma skin (Figure 4a). Furthermore, STX140 (100 nM) caused a significant decrease in IL-8 quantification only in SKMEL28-R, compared to the untreated control (Figure 4b). IL-6 was not affected by STX140 at the tested concentration.

### 2.2. STX140 Treatment Led to Cell Cycle Arrest

In order to study cell death induction caused by STX140 (Figure 5), we monitored the externalization of phosphatidylserine as a marker of early apoptosis which, combined with cell membrane fragmentation marked by positive nucleus fluorescence with propidium iodide (PI), indicated late apoptosis characteristics. Results showed that only 500 nM of STX140 was able to increase significantly (*p* < 0.05) the number of cells in early apoptosis in SKMEL-28P and -R, with no cells in late apoptosis.

Concerning cell cycle progression, although STX140 did not show remarkable apoptosis-triggering potential, this compound was able to cause cell cycle arrest at 200–500 nM concentrations with a significant increase (*p* < 0.001) in the number of cells in the G2/M phase compared to the untreated control in both parental and resistant melanoma cells (Figure 6). In addition, the significant increase (*p* < 0.05) in the percentage of cells in the sub-G1 population, commonly associated with DNA fragmentation, is also worth mentioning, caused by STX140 only at 500 nM and slightly more severe in SKMEL-28P.

### 2.3. STX140 Triggered Senescence Phenotype

Since parental and resistant melanoma cells suffered an antiproliferative activity with a cycle arrest in the G2/M phase and possible DNA damage with no apoptosis features, we evaluated the ability of STX140 to trigger senescence mechanisms. As evidenced in Figure 7a using the cell senescence dye, there was a significant increase (*p* < 0.01) concerning the untreated control by treatment with STX140 (100 nM), SA-β-gal stained in around 60% of SKMEL28-P and -R, by fluorescence microscopy. In addition, we observed melanoma cell morphology alterations when treated with STX140. These cells lost the characteristic stellate morphology into an acquired more spread-out shape.

Thus, associated with morphological and functional changes, the impact of STX140 on modulating the expression of genes related to senescence and apoptosis phenotypes was evaluated. As seen in Figure 7b, there was no alteration in the gene expression of SKMEL-28P. Also, *TP53*, *TP73*, *CDKN1B*, *CDKN2A*, *BBC3*, *PMAIP1*, *BAX*, and *SIVA1* genes remained unchanged in SKMEL-28R cell line. However, STX140 (100 nM) significantly (*p* < 0.05) increased *CDKN1A* and *GADD45A* expression in resistant melanoma cells.

Further, autophagy, apoptosis, and senescence signaling proteins were also quantified. STX140 could not significantly modulate the activity of PARP1 (apoptosis- and DNA damage-related) along with LC3B and SQSMT1/p62 (autophagy-related). However, compared to the negative control, STX140 induced significantly (*p* < 0.05) an elevated expression of p21 (only in SKMEL-28R) and γH2AX expression (in both SKMEL28-P and -R), as demonstrated in Figure 8.

## 3. Discussion

STX140, classified as a non-estrogenic steroidal bis-sulfamate, has shown antitumor therapeutic potential in vivo and in vitro reported in several studies [24]. STX140 has been shown to disrupt microtubule polymerization when interacting with tubulin leading to cell cycle arrest and apoptosis. In a mouse model of breast cancer, orally administered STX140 outperformed the chemotherapy drug paclitaxel, which also disrupts microtubule dynamics by reducing tumor size and preventing metastasis formation without showing neurotoxicity [29,30]. Also, STX140 showed efficacy when administered orally in both MCF-7^WT^ and MCF-7^DOX^ breast cancer xenograft models, in contrast to paclitaxel and 2-methoxyestradiol [23]. Currently, there are no clinical trials ongoing with STX140, but the compound has already shown many promising results in preclinical studies in oncology [31,32] and also elsewhere, in particular its recent in vivo activity on preventing experimental autoimmune encephalitis [33].

Considering in vitro models, since STX140 is a 2-ME analog, it is worth mentioning that our group previously reported the 2-ME activity against melanoma cells using a bi- and three-dimensional model. Thus, 2-ME inhibited the proliferation, clonogenicity, and invasive potential of BRAF^V600E^ mutated and BRAFi-resistant cells. In the potency evaluation assay, the IC_50_ for the SKMEL-28 cell line was 2.74 μM for 3 days, 6.81 μM for 5 days, and 4.41 μM for 7 days of treatment incubation [17]. As shown here, the 2-ME analog STX140 showed that it could reduce cell proliferation and clonogenicity potential at a nanomolar potency against melanoma cells that harbor BRAF^V600E^ point mutation and its BRAFi-resistant variation (Table 1).

In a study with ovarian, prostate, and breast cancer cells, STX140 inhibited tumor viability with IC_50_ values of 280, 260, and 250 nM, respectively [34]. These values corroborated those presented herein and showed slightly more sensitivity in melanoma cell lines to the effect of STX140, with lower IC_50_ values.

In addition, STX140 demonstrated an inhibition of the migration and invasion capacity of melanoma cells, mainly in SKMEL-28R, as seen in a wound-healing assay, reconstructed skin model, and inhibition of IL-8 expression. Analyzing the results of cell death characterization by Annexin-FITC and PI, we conclude that migration and invasion inhibition is not due to the induction of cell death since STX140 does not significantly induce death features (apoptosis/necrosis) in melanoma cells. Since other authors have shown that senescent cells commonly block intrinsic and extrinsic apoptosis signaling via antiapoptotic Bcl-2 family proteins [35,36,37], these results also contribute to the hypothesis that STX140 acts to prevent cell proliferation, migration, and invasiveness phenotype via senescence mechanisms.

Therefore, we evaluated if STX140 induces senescence features in melanoma, as senescence cells highly express the lysosomal enzyme β-galactosidase, classically marked positively using X-gal dye at pH 6 [38,39]. As mentioned before, STX140 led to positive blue staining to SA-β-gal and morphology characteristics such as a flattened cytoskeleton with a prominent nucleus and scattered cytoplasm. In addition to these parameters, quantitative PCR and blotting analysis demonstrated that STX140 upregulated the expression of the senescence gene and proteins related to senescence signaling. GADD45 stimulates proteins that have products that inhibit proliferation in tumor cells and, in response to different stresses, can induce senescence through interaction with p21 [40]. CDKN1A has p21 protein as a product, a potent cyclin-dependent kinase inhibitor that blocks cell cycle progression, and is a highly elevated protein in senescent cells in response to DNA injury. Furthermore, p21 is part of one of the p53/p21 signaling, a crucial main senescence pathway [41,42].

In a cell cycle evaluation, STX140 showed similar results compared to its parent 2-ME. 2-ME and STX140 demonstrated cell cycle arrest in the G2/M phase in human melanoma cells, but STX140 had activity in the nanomolar range. Although cell cycle arrest in G2/M was demonstrated as a microtubule inhibition consequence of 2-ME activity [17], it seems like STX140 is causing G2/M and sub-G1 arrest in response to DNA damage, triggering the senescent machinery activation. Studies have shown that other pharmacological compounds, such as aurora kinases inhibitors (AURK) and PLK1 inhibitors, can induce senescence features by stopping the passage of the G2 phase to the mitosis phase, consequently blocking the G2/M progression of the cell cycle via p21 and γH2AX [43,44].

Beyond the classical cell senescence induction through progressive telomere shortening in response to cell aging and genomic instability prevention, cells can also trigger an irreversible premature senescence signaling that can act as a tumor-suppressive function [45]. Double-stranded DNA breaks are powerful senescent signaling activators, usually inducing activation of H2AX aiming to recruit DNA damage response components [46]. DNA damage response also can act via the cyclin-dependent kinase inhibitor 1A (CDKN1A), triggering the p21 protein and resulting in cell cycle arrest. Even though these molecules related to cell cycle inhibition alone do not necessarily attempt to trigger senescence, these alterations together are an essential part of the hallmarks of the cellular senesce pathway [45,47,48,49,50]. In this context, STX140 activity modulated the expression of some of the hallmarks of the cellular senescence pathway, such as upregulating *CDKN1A* and *GADDA45* genes in addition to the elevation of the expression of γH2AX (a phosphorylated form of the histone variant H2AX) and p21 proteins, p53-independently, mostly seen in resistant human melanoma cells.

Moreover, although senescence signaling usually upregulates IL-6 and IL-8 activity in an autocrine manner part of the pro-inflammatory factors termed the senescence-associated secretory phenotype (SASP) [49,50,51], in melanoma cells, these interleukins are commonly associated with high tumor growth and invasiveness linked to poor prognosis [51,52]. Jobe et al. [53] showed that simultaneous blocking of IL-6 and IL-8 could represent a total inhibition of human melanoma invasiveness, which means that IL-8 downregulation could represent an STX140 anti-invasion pathway activity. Also, patients with a decrease in serum levels of IL-8 could reflect and predict response to anti-PD-1 treatment in melanoma and non-small cell lung cancer [54].

Even though the mechanism of action of STX140 was not tested entirely in non-melanoma cells, we analyzed STX140 cytotoxicity in the 3D reconstructed human skin model. The reconstructed human skin includes a group of samples with no melanoma (Figure 2a) consisting of three primary non-cancer cells (e.g., fibroblasts, keratinocytes, and melanocytes). This group exhibited no STX140 toxicity or morphological changes in the reconstructed human skin tissue.

Moreover, treatment with drugs that induce senescence and inhibit cell proliferation without triggering cell death currently presents an alternative to traditional cytotoxic therapies that can cause serious adverse effects in patients and often develop resistance [36]. The implication of using senotherapeutic drugs (compounds that modulate senescence) to treat cancer are being discussed currently and studied in various stages of development [55,56,57]. Also, further investigation should be necessary to understand the synergize potential of STX140 with other melanoma drug therapies used in clinics (such as BRAF and MEK inhibitors).

## 4. Materials and Methods

### 4.1. STX140 Synthesis, Cell Culture and Reagents

STX140 was synthesized according to the method of Leese et al. [58] (referred to as compound **21**).

Primary human melanocytes and fibroblasts were isolated from foreskin samples donated at the Hospital of the University of São Paulo (CEP/HU-USP 943/09, SISNEP CAAE 0062.0.198.000-9) as approved by the local ethics committee (CEP/FCF-USP 534). They were isolated and cultured according to the protocol previously described by our group [59].

Melanoma SKMEL-28 cell line (BRAF^V600E^ mutation) was kindly donated by Dr. Marisol Soengas (Centro Nacional de Investigaciones Oncológicas, Madrid, Spain). Melanoma cells with BRAF resistance phenotype were prior generated according to Sandri et al. [59] procedure. The parental and vemurafenib-resistant SKMEL-28 melanoma cell lines with BRAF^V600E^ mutation were used as a monolayer cell model. Cultivation was performed in DMEM (Dulbecco’s modified Eagle’s medium) for the SKMEL-28 cell supplemented with 10% fetal bovine serum (FBS) and antibiotics (25 µg/mL of ampicillin and 100 µg/mL of streptomycin). BRAFi-resistant SKMEL-28 cells were maintained in the presence of vemurafenib (3.0 μM, S1267, Selleck Chemicals, Houston, TX, USA). The cells were maintained in a humid incubator at 37 °C temperature under a 5% CO_2_ atmosphere. All cell lines were checked for mycoplasma contaminations by PCR assay.

### 4.2. Cell Viability Evaluation

Cell viability was accessed by trypan blue (0.4%, 1:1 culture medium) (1.11732, Merck, Temecula, CA, USA) dye exclusion assay. Cells were seeded (2 × 10^4^ cell/well) in a 24-well plate and treated with STX140 at 50 nM, 100 nM, 200 nM, and 400 nM concentrations for 24, 48, and 72 h of incubation. After dye addition, viable cells were counted in Neubauer chambers, considering colored cells as non-viable.

### 4.3. Clonogenic Assay

Cells were seeded in a 6-well cell culture plate (800 per well) and treated with STX140 at 25, 50, 100, and 200 nM concentrations twice a week during 10–17 days. After this period, cell colonies were stained with crystal violet (0.5% in a 1:1 water/methanol solution). The percentage of the area occupied by the colonies was quantified through a computational analysis.

### 4.4. Senescence Evaluation with β-Galactosidase Staining

Melanoma cells were seeded (2 × 10^4^ cells/well) in 6-well plates and treated with STX140 (100 nM) concentration for 48 h-incubation. Cells were fixated and stained of senescence-associated β-galactosidase (SA-β-gal) was performed using the Senescence β-galactosidase Staining Kit (9860, Cell Signaling Technology, Danvers, MA, USA) following the manufacturer’s instructions. Senescent and non-senescent cells were counted under the microscope in 6 different random fields for each group (treated and not treated).

### 4.5. Wound-Healing Assay

Melanoma cells were seeded (1 × 10^4^ cells/well) in a 24-well plate and left to reach approximately 90% confluence. Using a 200 μL tip, a vertical scratch was made in the center of each well. Then, cells were washed with phosphate buffered saline (PBS) and treated with STX140 (100 nM) diluted in DMEM culture medium + 1% FBS. The wound-healing process was measured by acquiring twice-per-well scratch images in time 0-, 12-, 24-, and 48-h incubation. Cell-free areas were calculated using Image J 3.7 software.

### 4.6. IL-6 and IL-8 Quantification

The supernatant of melanoma cell culture prior treated with STX140 (100 nM) was collected and frozen at −80 °C. Interleukins IL-6 (QC193, R&D Systems^®^, Minneapolis, MN, EUA) and IL-8 (R&D Systems^®^, AF-208-NA) from samples were quantified by ELISA assay (R&D Systems^®^) and analyzed according to the instructions of the commercial kit manufacturers. Optical density was immediately measured at 450 nm in a 96-well microplate spectrophotometer detector. Interleukin concentration was determined by interpolation from the standard curve.

### 4.7. Flow Cytometry—Apoptosis Evaluation

Cell lines were seeded in a 6-well plate (2 × 10^5^ cells/well) and treated with STX140 at 100, 200, and 500 nM concentrations in a 24-h incubation. After treatment, samples were washed twice and resuspended with ice-cold PBS, and stained with FITC Annexin V Apoptosis Detection Kit II (556570, BD Biosciences, Franklin Lakes, NJ, USA) according to the manufacturer’s instructions. Fluorescence was measured in a flow cytometer equipment (FACS Cantho, BD Biosciences). For interpretation, cells marked as both FITC Annexin V+/PI+ were considered either in the late stage of apoptosis; FITC Annexin V+/PI− were considered in the early stage of apoptosis; FITC Annexin V−/PI+ were considered already dead or necrosis; both FITC Annexin V−/PI− are alive and not undergoing measurable apoptosis.

### 4.8. Flow Cytometry—Cell Cycle

Cell lines were seeded in a 6-well plate (2 × 10^5^ cells/well) and treated with STX140 at 100, 200, and 500 nM concentrations in a 24-h incubation. Then, cell samples were washed with PSB solution and resuspended in a permeabilization and stain buffer (PBS containing 4 mg/mL of RNase A, triton x-100 0.1%, and 50 μg/mL of propidium iodide). After 30 min-incubation at room temperature in the dark, DNA quantification was measured by the sample’s fluorescence read in a flow cytometer (FACS Cantho, BD Biosciences).

### 4.9. Gene Expression Evaluation—RT-PCR

Total RNA from samples was extracted using TRIzol reagent (Thermo Fisher Scientific, Rockford, IL, USA). A total of 1 μg of RNA was reverse transcribed into cDNA with the High-Capacity cDNA Reverse Transcription Kit (Thermo Fisher Scientific). Real-time gene amplification was performed on the QuantStudio 3 Real-Time PCR System equipment and the Power SybrGreen reagent (Thermo Fisher Scientific) with specific primers (Table 2). Fifteen nanograms of each cDNA sample were used in the reactions and a negative control, with the addition of water (nuclease-free) in place of cDNA and performed for each pair of primers. HPRT1 and ACTB were used as reference genes. The relative expression was calculated using the formula 2^−ΔΔCT^ (PMID:11846609) [60].

### 4.10. Reconstructed Human Skin (RHS)

The reconstructed human skin model was based on the previous description in Brohem et al. [61] with minor changes. In summary, dermis equivalent was constructed in a 6-well plate with fibroblast cells (1.5 × 10^5^ cells/well) plus collagen type I (BD Biosciences). Melanoma cells (5 × 10^5^ cells/well) were incorporated into the epidermis along with keratinocytes and melanocytes (2.5 × 10^5^ and 0.8 × 10^4^ cells/well, respectively). After 24 h-submersion, skin samples were transferred to the air-liquid interface and maintained for 10 days with DMEM/HAM’s F-12 supplemented culture medium with 10% FBS. Then, 24 h prior to tissue collection and fixation (4% paraformaldehyde for 4 h and 70% alcohol for 24 h), RHS were treated with STX140 (500 nM). Tissue samples were stained with hematoxylin/eosin (H&E) for histological analysis.

### 4.11. Statistical Analysis

Data analysis was carried out using GraphPad Prism 8.0 software. Values are expressed as the mean ± standard deviation (S.D.) of three independent experiments (n = 3). Statistical significance (significant when *p* < 0.05) was analyzed using one-way analysis of variance (ANOVA).

## 5. Conclusions

This study demonstrates that STX140 has potent antiproliferative and anti-invasive effects in the nanomolar range on BRAFi-resistant melanoma cells while having a less pronounced effect on human melanocytes. Additionally, STX140 induced cell cycle arrest in the G2/M phase and sub-G1, reduced migration, and clonogenic potential in monolayer models, and inhibited invasion in a three-dimensional human skin model with melanoma cells. Moreover, STX140 induced senescence features in melanoma and activated the senescence machinery by upregulating the expression of senescence genes, such as CDKN1A e GADD45A in resistant melanoma and proteins related to senescence signaling, such as p21 (only in SKMEL-28R) and γH2AX. Preclinical studies conducted on STX140 demonstrate its potential as a therapeutic agent for the treatment of other cancer types, and clinical trials are necessary to confirm its safety and effectiveness in humans. However, STX140’s significant cytostatic and antiproliferative effects underline the need for continued research on this promising compound.

## Figures and Tables

**Figure 1 ijms-24-11314-f001:**
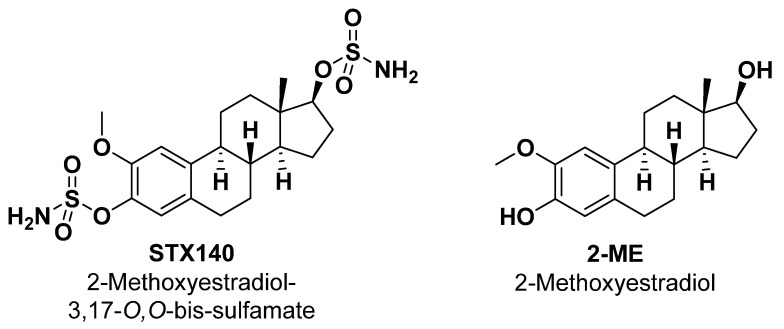
Chemical structures of STX140 and 2-ME.

**Figure 2 ijms-24-11314-f002:**
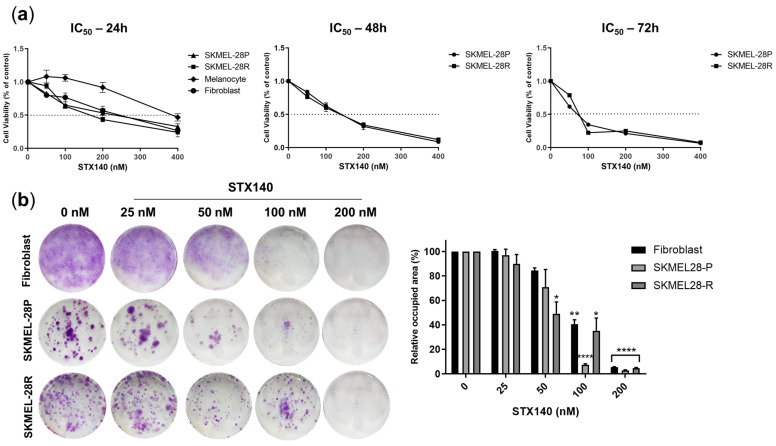
Antiproliferative activity of STX140 against human melanoma cells. (**a**) Cell viability evaluation of melanoma and primary cells for IC_50_ estimation; (**b**) clonogenic assay. Results are expressed as the mean of three independent experiments (*n* = 3) ± S.D.; significance is indicated by * *p* < 0.05, ** *p* < 0.01, **** *p* < 0.0001. Representative micrographs from inverted microscope (4×).

**Figure 3 ijms-24-11314-f003:**
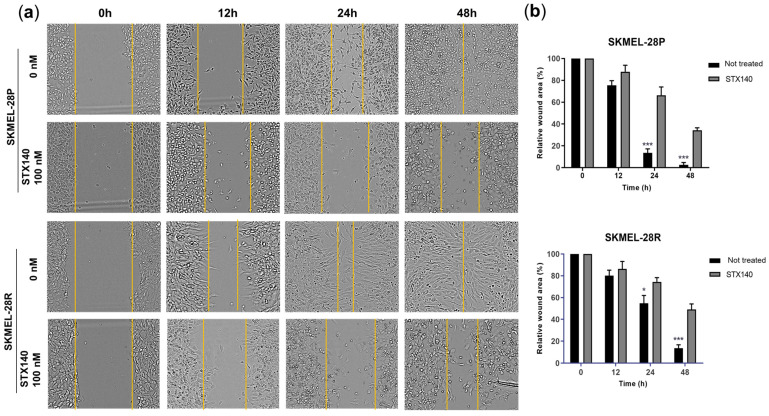
Antimigration activity of STX140 against human melanoma cells. (**a**) Representative micrographs from inverted microscope (20×); (**b**) Graphical representation, results are expressed as mean of three independent experiments (*n* = 3) ± S.D.; significance is indicated by * *p* < 0.05 and *** *p* < 0.001. Yellow lines represent the edge of the cell-wound.

**Figure 4 ijms-24-11314-f004:**
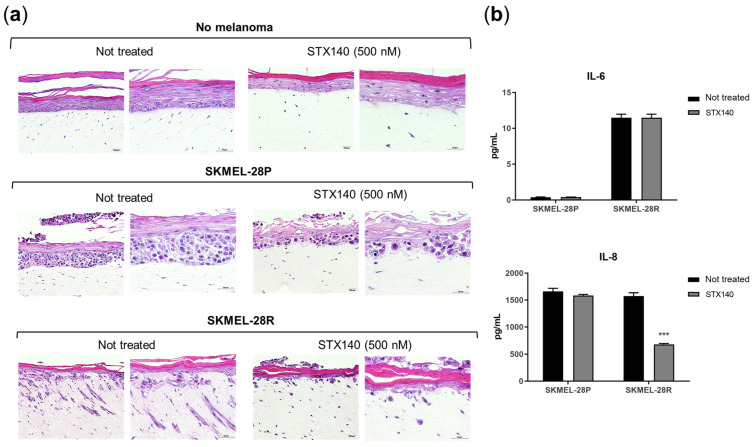
Anti-invasion activity of STX140 against human melanoma cells. (**a**) Reconstructed human skin with melanoma micrographs from inverted microscope (20× and 40×, respectively) dyed with hematoxylin/eosin (H&E); (**b**) Interleukin quantification in media from the 3D reconstructed human skin samples, STX140 (100 nM), results are expressed as the mean of three independent experiments (*n* = 3) ± S.D.; significance is indicated by *** *p* < 0.001.

**Figure 5 ijms-24-11314-f005:**
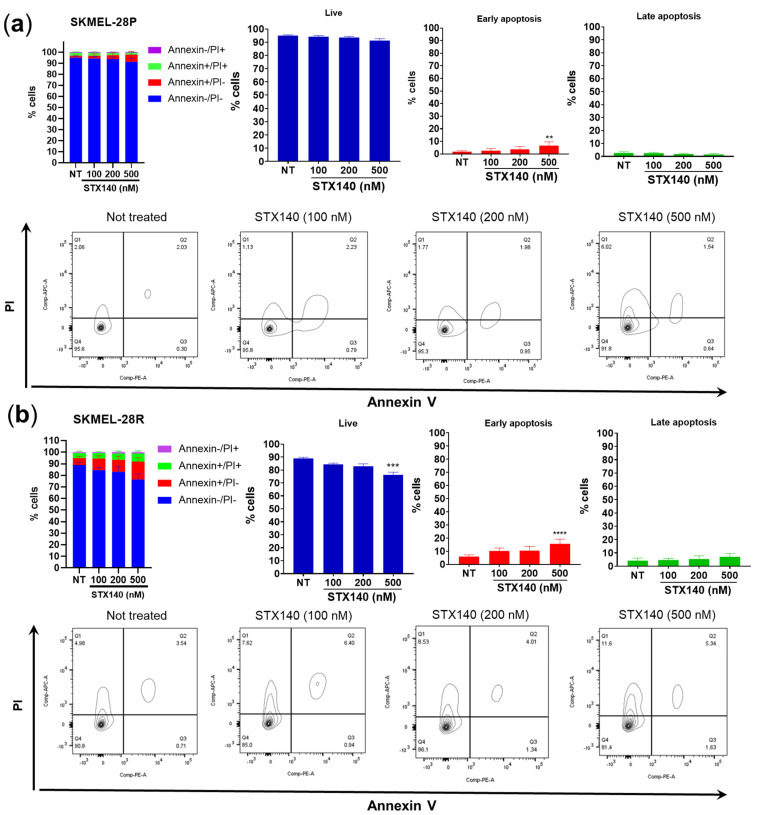
Cell death evaluation by STX140 activity against human melanoma cells. (**a**) SKMEL-28P; (**b**) SKMEL-28R. Graphical results are expressed as the mean of three independent experiments (*n* = 3) ± S.D.; significance is indicated by ** *p* < 0.01, *** *p* < 0.001, and **** *p* < 0.001. PI = propidium iodide; NT = not treated cells.

**Figure 6 ijms-24-11314-f006:**
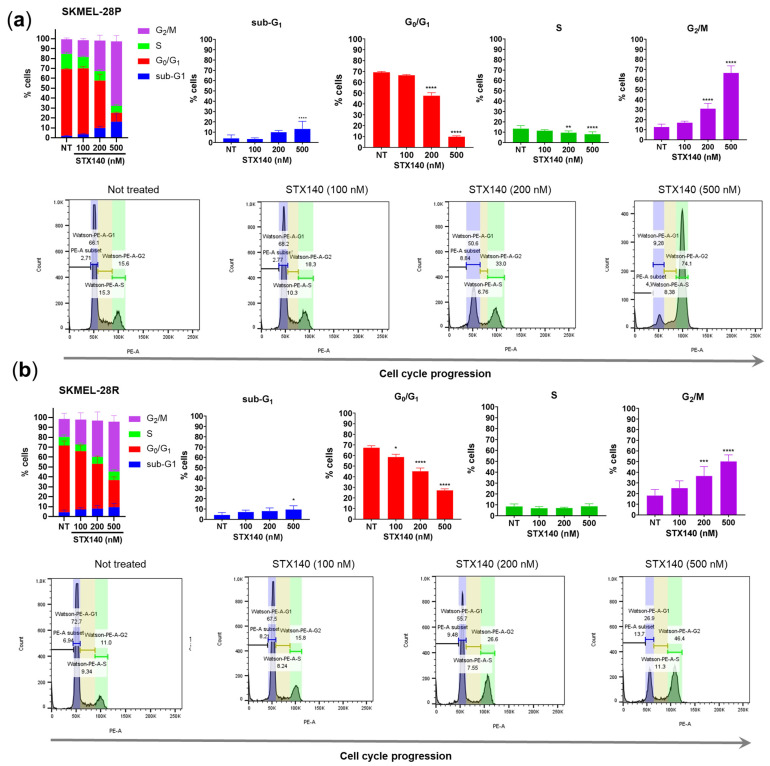
Cell cycle evaluation by STX140 activity against human melanoma cells. (**a**) SKMEL-28P; (**b**) SKMEL-28R. Graphical results are expressed as the mean of three independent experiments (*n* = 3) ± S.D.; significance is indicated by * *p* < 0.05, ** *p* < 0.01, *** *p* < 0.001, **** *p* < 0.0001. NT = not treated cells.

**Figure 7 ijms-24-11314-f007:**
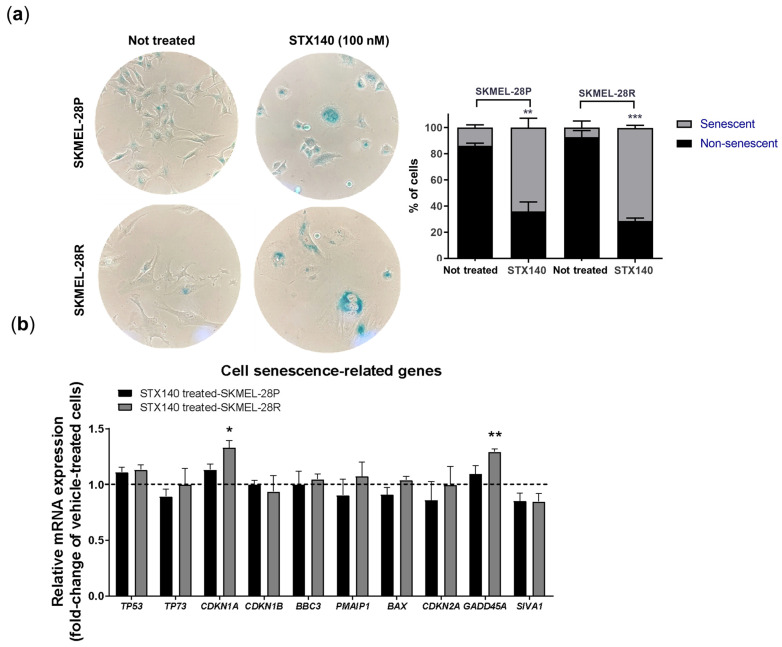
STX140 (100 nM) treatment induces senescence profile in human melanoma cells. (**a**) β-galactosidase staining with statistical quantification; (**b**) Cell senescence-related genes modulation; Results are expressed as the mean of three independent experiments (*n* = 3) ± S.D.; relative mRNA expression is compared to not treated cells (represented as the dash lines), significance is indicated by * *p* < 0.05, ** *p* < 0.01, and *** *p* < 0.001. Representative micrographs from inverted microscope (10×).

**Figure 8 ijms-24-11314-f008:**
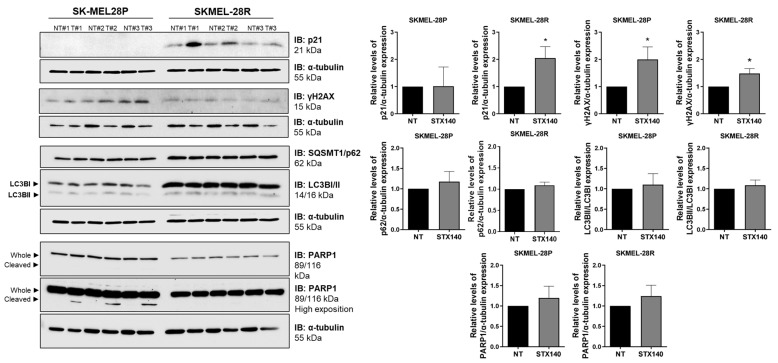
Cell death and senescence-related signaling modulation by STX140 (100 nM) activity in human melanoma cells. Results are expressed as mean of three independent experiments (*n* = 3) ± S.D.; significance is indicated by * *p* < 0.05. NT = no treated cells.

**Table 1 ijms-24-11314-t001:** STX140 antiproliferative activity quantification.

	SKMEL-28P	SKMEL-28R	Melanocyte	Fibroblasts
24 h	95.3	112.3	222.3	133.5
48 h	114.9	101.0	ND	ND
72 h	61.9	68.2	ND	ND

Half-maximal inhibitory concentration (IC_50_) values are expressed in nanomolar (nM). ND = not determined.

**Table 2 ijms-24-11314-t002:** Primers sequences and gene functions.

Gene	Sequence	Concentration
*TP53*	FW: GGCGCACAGAGGAAGAGAAT RV: GGAGAGGAGCTGGTGTTGTTG	300 nM
*TP73*	FW: GCACCACGTTTGAGCACCTCT RV: GCAGATTGAACTGGGCCATGA	300 nM
*CDKN1A*	FW: TGTCACTGTCTTGTACCCTTGT RV: GCCGGCGTTTGGAGTGGTAG	300 nM
*CDKN1B*	FW: ACTCTGAGGACACGCATTTGGT RV: TCTGTTCTGTTGGCTCTTTTGTT	300 nM
*BBC3*	FW: GACCTCAACGCACAGTACGAG RV: AGGAGTCCCATGATGAGATTGT	300 nM
*PMAIP1*	FW: CGCGCAAGAACGCTCAACC RV: CACACTCGACTTCCAGCTCTGCT	300 nM
*BAX*	FW: GAGCTGCAGAGGATGATTGC RV: CAGCTGCCACTCGGAAAA	300 nM
*CDKN2A*	FW: CACCGAATAGTTACGGTCGGA RV: CACGGGTCGGGTGAGAGTG	300 nM
*GADD45A*	FW: AAGGATGGATAAGGTGGGG RV: CTGGATCAGGGTGAAGTGG	300 nM
*SIVA1*	FW: TCTTCGAGAAGACCAAGCG RV: TGCCCAAGGCTCCTGATC	300 nM
*ACTB*	FW: AGGCCAACCGCGAGAAG RV: ACAGCCTGGATAGCAACGTACA	150 nM
*HPRT1*	FW: GAACGTCTTGCTCGAGATGTGA RV: TCCAGCAGGTCAGCAAAGAAT	150 nM

## Data Availability

Data are available upon request.

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
