# Peer review of "2-Methoxyestradiol-3,17-O,O-bis-sulfamate (STX140) Inhibits Proliferation and Invasion via Senescence Pathway Induction in Human BRAFi-Resistant Melanoma Cells"

_ijms, 2023, doi:10.3390/ijms241411314_

Round 1

Reviewer 1 Report

This is a very interesting manuscript by Ylana Adami Franco et.al, explaining detailed mechanism of 2-methoxyestradiol-3,17-O,O-bis-sulfamate (STX140) inhibition of melanoma cells. Authors also proposed STX 140 as a therapeutic target to treat melanoma cancer. Overall, the manuscript was well written with report data was good with figures. Below are my comments.

Lane 90-91: Why STX140 concentration >100nM was used in migration, anti-invasion, cell death analysis and in other experiments.

Lane 111-113: representing results corresponding to figures legends is missing.

Lane 116: what are yellow lanes in figure 3a?

Lane 121: Why there is difference in inhibition of secretion of IL-6 and IL-8 (fig 4b) and label the 4a figure.

Lane 169-170: Relative gene expression is compared to untreated control?

Reviewer 2 Report

Overall, an important topic of investigating new therapeutics to address drug resistance in melanoma. The data is well organized, presented and clearly discussed. 

Testing STX140 on single parental and resistant cell line does not seem sufficient due to the heterogeneity of resistance. Could another parental and resistant cell line be included?

The effects of the STX140 on non-cancer cells is not examined in the majority of the assays. Could control cells be included in at least one or two additional assays to understand the effects on non-cancer cells?

The evidence of senescence induction is not very strong. Senescence is not typically confirmed immediately after a 48 hour drug treatment as in Figures 7-8. Typically, drug treatment is used to induce senescence for 48-72 hours then there is a recovery phase of a 3-5 days post-drug treatment. This is used to confirm that it is not a transient cell stress response and that the cells cannot recover and start proliferating once the drug is removed. Is the STX140 induced senescence maintained after the drug is removed? Does additional time for senescence induction increase senescence markers?

How does this treatment compare and/or synergize with other melanoma drug therapies (BRAFi, MEKi, chkpt inhibitors)? Could any of these be tested?

Minor comments:

Figure 2 a) why was toxicity in fibroblasts and melanocytes not tested at 48 and 72 hours? b) The fibroblasts are over-confluent and not at a proper cell dilution to form independent colonies.

Figure 3 a) "Control" implies non-cancer cells, instead label as untreated or not treated (NT) as used in Figure 6.

Figure 4 b) Please clarify this was protein quantification in media for IL6 and IL8 in the figure legend and text. Please clarify that it was not human skin samples but 3D skin models. While it is explained in the discussion, a brief description of relevance of IL6 and IL8 testing in the results section would have been helpful.

Figure 5 is incorrectly labeled because early apoptosis and necrosis are switched in all the graphs based on the FACS plots. Doxorubicin induced primarily early apoptosis (annexin V+ and PI-) not necrosis (annexin V- and PI+), and STX140 did not induce early apoptosis, but necrosis at 500 nM. Please correct figure and text.

Figure 6 what does STX140 do to the cell cycle in non-cancer cells?

Figure 8 SKMEL-28R p62 graph mislabeld with p21

restructure sentence 133-135

268 liked to linked

255 gammaH2AX 
